# A Fine-Grained Semantic Alignment Method Specific to Aggregate Multi-Scale Information for Cross-Modal Remote Sensing Image Retrieval

**DOI:** 10.3390/s23208437

**Published:** 2023-10-13

**Authors:** Fuzhong Zheng, Xu Wang, Luyao Wang, Xiong Zhang, Hongze Zhu, Long Wang, Haisu Zhang

**Affiliations:** College of Information and Communication, National University of Defense Technology, Wuhan 430074, China; zhengfuzhong21@nudt.edu.cn (F.Z.); wangxu9191@gmail.com (X.W.); wangluyao@nudt.edu.cn (L.W.); zhangxiong17@nudt.edu.cn (X.Z.); 18062669006@163.com (H.Z.); wanglong17c@nudt.edu.cn (L.W.)

**Keywords:** multi-scale, fine-grained semantic alignment, remote sensing, cross-modal retrieval

## Abstract

Due to the swift growth in the scale of remote sensing imagery, scholars have progressively directed their attention towards achieving efficient and adaptable cross-modal retrieval for remote sensing images. They have also steadily tackled the distinctive challenge posed by the multi-scale attributes of these images. However, existing studies primarily concentrate on the characterization of these features, neglecting the comprehensive investigation of the complex relationship between multi-scale targets and the semantic alignment of these targets with text. To address this issue, this study introduces a fine-grained semantic alignment method that adequately aggregates multi-scale information (referred to as FAAMI). The proposed approach comprises multiple stages. Initially, we employ a computing-friendly cross-layer feature connection method to construct a multi-scale feature representation of an image. Subsequently, we devise an efficient feature consistency enhancement module to rectify the incongruous semantic discrimination observed in cross-layer features. Finally, a shallow cross-attention network is employed to capture the fine-grained semantic relationship between multiple-scale image regions and the corresponding words in the text. Extensive experiments were conducted using two datasets: RSICD and RSITMD. The results demonstrate that the performance of FAAMI surpasses that of recently proposed advanced models in the same domain, with significant improvements observed in R@K and other evaluation metrics. Specifically, the mR values achieved by FAAMI are 23.18% and 35.99% for the two datasets, respectively.

## 1. Introduction

The rapid advancement of remote sensing technology [1] has led to the widespread application of remote sensing (RS) images in various domains, including vegetation analysis [2,3], geological research [4], urban planning [5], disaster early warning [6,7], and earth observation [8]. Consequently, significant progress has been made in these areas of research. However, the scale of remote sensing data has also increased considerably [9]. Dealing with the vast amount of multi-source and multi-temporal remote sensing images [10] presents a significant challenge [11,12,13], necessitating the improvement of data processing efficiency and intelligence. In recent years, with the widespread application of deep learning technologies, content-based image retrieval (CBIR) has made significant progress, continually improving its accuracy [14,15,16,17]. However, this uni-modal retrieval approach, which involves searching from RS image to RS image, significantly restricts the flexibility of user queries. Individuals frequently favor semantic-based image retrieval over the search for visually similar images. Given the flexibility of text in human–computer interactions, researchers have recently shown interest in RS cross-modal text-image retrieval (RSCTIR) [18].

Recent studies on similarity measurement in cross-modal retrieval can be categorized into two main methods: those based on a common feature space and those based on fusion coding. In the former approach, the cosine distance or Hamming distance is commonly utilized. The latter approach transforms the distance metric into a binary classification problem to determine whether a match exists. In the common feature space method, Abdullah et al. [19] were pioneers in extending the cross-modal embedding technique from natural images to remote sensing. They proposed a deep bidirectional triplet network to generate embedded representations of both images and text, facilitating similarity measurement between them. Rahhal et al. [20] employed an image-text contrastive learning approach along with InfoNCE loss to ensure high similarity between paired image-text features. Given the complexities inherent in remote sensing image application scenarios and objectives, Cheng et al. [21] introduced an attention mechanism and a gate mechanism to enhance the fine-grained correspondence between image regions and words, simultaneously filtering out redundant information. Another study by Cheng et al. [22] introduced a multi-attention fusion module to mitigate background noise interference in remote sensing images and emphasize salient objects. Yuan et al. [23] proposed a GaLR model that facilitated the mutual correction and supplementation of global and local information through multi-level information dynamic fusion. From an image-centric perspective, Yu et al. [24] heightened the interaction between intra-modal features using graph neural networks, treating image regions (or text words) as nodes within the image. Additionally, other aspects of RSCTIR have also been explored. For instance, Yuan et al. [25] adapted a lightweight model design from previous work [18] using knowledge distillation. Furthermore, Alsharif et al. [26] conducted cross-language and cross-modal retrieval for the first time, investigating the joint application of English and Arabic based on a transformer model. In terms of retrieval efficiency for real-valued features, Mikriuko et al. [27] proposed a novel unsupervised contrastive hashing network that trained hash modules using multi-target losses, resulting in high-quality hash coding.

As for the fusion coding method, Li et al. [28] introduced the contrastive loss to enforce alignment between single-modal features. They further employed a three-layer transformer encoder as the fusion encoder to explore the potential correspondence between the contrastive loss and alignment. Lv et al. [29] adopted a Siamese network to obtain the joint representation of image–text pairs. They improved the quality of the joint representation by leveraging both cross-entropy loss and center loss. Subsequently, they employed L2 loss to align the single-modal features with the fusion feature, effectively transferring knowledge from the fusion encoder to the single-modal encoders. This approach presents a novel concept of combining the fusion coding method with the common feature space method.

Furthermore, several studies address the challenges posed by the multi-scale features of remote sensing images, as the differences in target scales make the semantic alignment of cross-modal features more complex [30]. As documented in [18,24,30,31], two main challenges arise in cross-modal retrieval due to multiple scales: (1) effectively utilizing the diverse scale features of an image, including emphasizing salient features and preserving information related to small targets; (2) modeling the intricate relationships among multi-scale targets. To address these challenges, Yuan et al. [18] introduced a multi-scale vision self-attention module that comprehensively investigates multi-scale information and eliminates redundant features by merging cross-layer features of a convolutional neural network (CNN). Additionally, Wang et al. [31] designed a lightweight sub-module for multi-scale feature exploration that utilizes parallel networks with distinct receptive fields to extract and integrate multi-scale features. Yao et al. [30] focused on modeling the relationships among multi-scale targets by constructing hypergraph networks at different levels to depict the connections between objects of varying scales.

While the aforementioned studies have made significant progress in addressing the challenges associated with multi-scale features, there is still room for further improvement. Firstly, the existing methods primarily focus on extracting and modeling of multi-scale features, without adequately considering the semantic alignment with text. In the context of describing remote sensing images, it is crucial to establish semantic associations between different scales of targets and accurately associate them with specific words (Figure 1). Secondly, the current methods often overlook the importance of ensuring consistency across layers during the fusion of cross-layer features. Shallow features contain valuable spatial information but possess weaker semantic discrimination abilities, whereas deep features exhibit stronger semantic discrimination capabilities [32].

To address the challenges above, this study presents a novel method that combines multi-scale feature representation of images with fine-grained (between image regions and words) cross-modal semantic alignment. The proposed method consists of two key components: (1) Multi-scale feature modeling and feature consistency enhancement, and (2) fine-grained semantic alignment. In the initial component, we utilize the DetNet-59 [33] as the backbone network for extracting image features, capitalizing on its modest downsampling factor to retain information pertinent to small targets. Following this, we employ a computation-friendly cross-layer feature connection method to construct multi-scale feature representations of the image. This method effectively captures the rich spatial information in shallow features and retains details of small targets during downsampling [34]. To address the semantic inconsistency of cross-layer features, an effective cross-layer feature consistency enhancement module (FCEM) is designed. In the second component, we generate the multi-scale features by adopting the idea of spatial pyramid pooling (SPP) [35]. Each scale’s feature map is individually mapped into blocks and then concatenated. Next, the concatenated feature maps, along with the word feature, are fed into a shallow cross-attention network. This network facilitates fine-grained semantic interaction between the multi-scale image features and the corresponding words in the text. By combining these two components, the proposed FAAMI method aims to achieve comprehensive multi-scale feature modeling and precise semantic alignment between image regions and text words.

The main contributions of this paper are as follows:

1. We introduce FAAMI (Fine-grained semantic Alignment and Aggregation of Multiscale Information) to tackle the challenge of aligning multi-scale features with textual information. This approach adeptly amalgamates the merits of multi-scale techniques and fine-grained semantic alignment methods. Firstly, FAAMI incorporates multi-scale fields of view, overcoming the limitation of traditional fine-grained semantic alignment networks that may overly fragment large-scale targets. Secondly, it establishes the semantic association between multi-scale features and words through fine-grained semantic alignment, thereby leveraging the full potential of multi-scale information.

2. To address the issue of feature inconsistency arising from the cross-layer feature connection method, we introduce FCEM. This module consists of a channel attention module based on second-order pooling and a self-attention module utilizing a transformer framework. The channel attention module enhances the semantic consistency of cross-layer features in terms of channel dimension, while the self-attention module improves consistency in the spatial dimension. Additionally, the module effectively filters out internal associations between redundant image information and the multi-scale features of the modeled image.

The remaining sections of this paper are shown below:The Section 2 provides an overview of recent studies concerning methods for constructing multi-scale features. The Section 3 elaborates on the proposed method in a comprehensive manner. In the Section 4, we present and scrutinize the experimental outcomes. The Section 5 delves into the significance of maintaining a well-balanced approach in multi-modal encoders for cross-modal retrieval models. Finally, in the Section 6, we summarize the findings of this study.

## 2. Related Work

In addition to cross-modal RS image retrieval, the issue of scale also affects various computer vision tasks, including image classification, object detection, and semantic segmentation. Regardless of the specific task, addressing the scale problem requires the establishment of image feature representations in different scale spaces. In this section, we review three multi-scale feature construction methods based on CNN frameworks (Figure 2). These methods served as references for our study.

### 2.1. Image Pyramid

The image pyramid technique is a straightforward approach to constructing multi-scale feature representations [36]. This method employs Gaussian kernel convolution and downsampling to create a Gaussian pyramid, which consists of a series of images and emulates the varying scale space at the network input. Both hand-crafted feature-based methods [37,38] and deep learning-based methods [39,40] have utilized the image pyramid to mitigate scale-related challenges in various tasks. However, within a deep network framework, all network layers are required to provide feature responses to the pyramid input. This significantly increases computational and memory overhead during training and inference, limiting the practical applicability of the image pyramid. Recent enhancements have predominantly concentrated on enhancing the efficiency of algorithms for particular tasks. For example, during target detection, ref. [41] introduced the SNIPER model, which partitions the image into a grid during training to diminish the training overhead. Additionally, ref. [42] reduced the expenses associated with both training and inference by employing reinforcement learning to ascertain the sequence in which image regions are scrutinized, transitioning from coarser to finer levels.

### 2.2. Parallel Network Branches

This method typically utilizes a specific layer of the CNN feature map as a starting point to concurrently construct multi-scale feature representations of an image using convolution kernels of different sizes. In a study by He et al. [35], the significance of scale problems in deep networks was established, and spatial pyramid pooling layers were proposed as an alternative to standard global pooling. This approach employed different partition pooling methods, such as dividing the image into different bins (e.g., 2 × 2, 3 × 3, and 4 × 4), to create distinct receptive fields and simulate the scale effect. In [43], Chen et al. introduced the ASSP (Atrous Spatial Pyramid Pooling) method, which replaced spatial pooling with parallel computations of dilated convolution kernels with varying dilation coefficients. Similar ideas were employed in [32,44,45,46], where multi-scale feature representations were constructed by considering receptive fields as variables across different branches. Nevertheless, these investigations introduced innovative enhancements in the configuration of receptive fields or the consolidation of information from various branches. The parallel branch method has demonstrated its effectiveness in constructing multi-scale representations. Although it incurs additional computation overhead, this branch method is commonly used only at the end of the backbone network’s operations. Notably, [32] proposed the use of trident blocks with parallel branches in place of residual blocks in the backbone network, with shared weights among the branches. This encouraged the network to learn scale-invariant features more effectively. During the detection stage, only one branch is utilized, reducing the computation overhead caused by multiple branches.

### 2.3. Cross-Layer Feature Connection

This method can be viewed as a compromise between the two aforementioned methods. The image pyramid method and the parallel branch method share a similar “parallel” concept, with the former applied to the network input and the latter utilized in the network structure. In contrast, the cross-layer feature connection method employs a typical “serial” approach and approximates the scale features based on interlayer-receptive fields of the CNN [47]. The efficacy of this approach was initially showcased in [48]. Furthermore, this method optimally leverages existing computational outcomes, resulting in minimal additional computational overhead and a computationally efficient nature. However, this characteristic also gives rise to disparities in cross-layer features, such as the contrast between abundant shallow spatial information and potent deep semantic discrimination capability. The Feature Pyramid Network(FPN) [49] stands out as a renowned framework designed to tackle this concern. FPN improves the consistency of interlayer features through a step-by-step upsampling process and linear summation, facilitating information fusion and interaction across layers to enhance feature consistency. Building upon FPN, numerous researchers have explored methods of improving the fusion of cross-layer features further. In [50], Liu et al. proposed the PANet framework, which introduced fusion paths from shallow to deep layers to enhance spatial information representation. Kong et al. [51] and Pang et al. [52] utilized different approaches to calculate a unified fusion feature, which was then combined with the feature pyramid layer by layer for reconstruction. Tan et al. [53] employed complex interlayer connection methods by repeatedly stacking feature connection structures to enhance feature representation. Moreover, Ghaisi et al. [54] incorporated reinforcement learning into the fusion network, allowing for the adaptive selection of feature layers for fusion without manual intervention. In summary, prior research on the cross-layer feature connection method predominantly concentrates on fusion techniques aimed at alleviating disparities in cross-layer features.

### 2.4. Summary of the Literature

Out of the three discussed methods for constructing multi-scale features, the image pyramid is the most straightforward, but it comes with significant computational and memory overheads. On the other hand, both the cross-layer feature connection method and the parallel branch method aim to approximate the image pyramid by generating different receptive fields within the network. However, they differ in their trade-offs. The cross-layer feature connection method incurs minimal computation overhead and retains rich spatial information from shallow features, but it can result in inconsistent cross-layer features. On the contrary, the parallel branch method ensures feature consistency but involves higher computational costs. The downsampling factor of the multi-scale feature maps in the parallel branch method is often large, leading to insufficient spatial information being retained.

Apart from the conventional CNN framework, researchers have also developed methods to address scale-related challenges in deep networks based on vision transformer structures. For example, Ren et al. [55] introduced the Shunted Self-Attention technique, which simulated the scale effect by modeling objects of different scales using separate attention heads within the same layer. They also adaptively aggregated multi-scale tokens, yielding promising results. However, when selecting a method for multi-scale feature construction, it is essential to consider the specific requirements of the task at hand. The chosen method should ideally complement the task’s objectives as much as possible.

## 3. Methodology

This section provides a detailed description of the proposed FAAMI. The framework of FAAMI is illustrated in Figure 3. Firstly, the initial embedded representation was obtained using single-modal encoders for the image and text (referred to as modules A and B in Figure 3). Transfer learning based on a pre-trained model [56] has proven to be exceptionally effective in this endeavor, facilitating the extraction of single-modal feature representations with a high degree of semantic discrimination. Additionally, we obtained global, high-level, and low-level semantic features by establishing cross-layer connections, and for the first time, we improved feature consistency using a channel attention mechanism grounded in second-order pooling. Furthermore, we incorporated the concept of the parallel branch method: features of varying scales were transformed into region feature sequences based on particular receptive fields and subsequently interconnected to form a comprehensive multiscale image representation. Furthermore, a three-layer self-attention mechanism (module C in Figure 3) was employed to strengthen feature consistency. Finally, the region features of the image and the word features of the text were input into the shallow cross-attention network (module D in Figure 3) to enable fine-grained semantic interaction. Notably, a suitable temperature control function was designed to regulate the operations of the cross-attention network, allowing the network to focus on samples with varying degrees of difficulty during different training stages.

In this section, we introduce modules A and B in Section 3.1, module C in Section 3.2, and the temperature control function and module D in Section 3.3.

### 3.1. Single-Modal Embedding

The primary task at hand is selecting a high-quality backbone network for image coding. To ensure the preservation of small target semantics while maintaining a lightweight model, we opted for DetNet-59, which is known for its excellent performance in target detection. Compared to ResNet [57], DetNet has a downsampling factor of 16, which is smaller. In terms of parameters, DetNet falls between ResNet-50 and ResNet-101. In this study, we saved the DetNet inputs at each stage and passed them to the FCEM for further processing, enabling the construction of multi-scale feature representations. Letting the image set be $M=\{m_{1},m_{2}\ldots \ldots ,m_{N}\}$ and $\forall m_{i}\in M$, the image feature extraction process of CNN in each stage can be expressed as:(1)$$\begin{array}{c} {\left\{f_{v}^{s}\right\}}_{s=1}^{6}=CNN\left(m_{i},\theta_{m}\right) \end{array}$$

$f_{v}^{s}$ represents the DetNet inputs of an image *v* in different stages. For example, when s=1, $f_{v}^{s}$ represents the image features of the first stage. DetNet has six stages.

Furthermore, the selection of an appropriate text encoder to extract textual features is imperative. In the FAAMI method, the pre-trained Bert [58] model was chosen for this purpose.

The text coding process of the Bert network can be recorded as shown below simply given that the text set is $T=\{t_{1},t_{2},\ldots \ldots ,t_{N}\}$ and $\forall t_{i}\in T$, where *t* is a word set $t_{i}=\left\{w_{1},w_{2},\ldots \ldots ,w_{p}\right\}$, and *p* is the number of words in this text:(2)$$\begin{array}{c} f_{t}=\left\{e_{clc},e_{1},\ldots ,e_{p}\right\}=Bert\left(t,\theta_{t}\right) \end{array}$$

$f_{t}$ represents the feature vector of a text extracted via the Bert network and is a set of word feature vectors $e_{i}\in R^{h}$. In general, the Bert network would add a classification head $e_{clc}$ at the beginning of a sentence; hence, $f_{t}\in R^{(p+1)\times h}$ where *h* is the feature dimension of each word.

### 3.2. Feature Consistency Enhancement Network

#### 3.2.1. Feature Cascading

Following extraction from all stages of the CNN, the image features were forwarded to the feature consistency enhancement network to generate multi-scale region feature sequences, as illustrated in Figure 4. To be more precise, we concatenated the image feature outputs from the first three stages of the CNN to derive the low-level semantic features, as illustrated below:(3)$$\begin{array}{c} f_{v}^{low}=Cat\left(downsample\left(f_{v}^{1}\right),f_{v}^{2},upsample\left(f_{v}^{3}\right)\right) \end{array}$$

In the equation above, “Cat” denotes the concatenation of feature matrices along the channel dimension. The “downsample” operation refers to average pooling, while “upsample” corresponds to bilinear interpolation. Both “downsample” and “upsample” operations aim to ensure that the feature maps have the same sizes.
(4)$$\begin{array}{c} f_{v}^{high}=f_{v}^{4}⊙\alpha +f_{v}^{5}⊙\left(1-\alpha \right) \end{array}$$
(5)$$\begin{array}{c} \alpha =\sigma (\omega_{1}Cat\left(f_{v}^{4},f_{v}^{5}\right)) \end{array}$$

The feature outputs from the fourth and fifth stages of the CNN were linked through a gate function to acquire the high-level semantic features, as demonstrated in Formula (4). Employing the gate function, rather than the cascading method, helps in diminishing the number of feature channels in subsequent calculations. ⊙ represent the multiplication of elements corresponding to the matrix. Formula (5) represents the operation process of the gate function. α is determined by the joint features of the fourth and fifth stages; σ is a sigmoid function; $\omega_{1}$ is a learnable parameter matrix: $\omega_{1}\in R^{C_{2}\times 2C_{2}}$ where $C_{2}$ represents the number of channels of $f_{v}^{4}$ and $f_{v}^{5}$.
(6)$$\begin{array}{c} f_{v}^{\mathrm{glb}}=f_{v}^{6} \end{array}$$

The outputs from the sixth stage of the CNN were considered as the global semantic features and did not undergo any further computations. Therefore, the dimensions of all the computation results are $f_{v}^{low}\in R^{C_{1}\times H_{1}\times W_{1}}$, $f_{v}^{high}\in R^{C_{2}\times H_{2}\times W_{2}}$, and $f_{v}^{glb}\in R^{C_{2}\times H_{2}\times W_{2}}$, respectively, where *C*, *H*, and *W* represent the sizes of a feature map in three dimensions (i.e., channel, length, and width).

#### 3.2.2. Channel Attention Based on Global Second-Order Pooling

This method was introduced by Gao et al. in [59]. It differs from the SENet [60] approach, which utilizes first-order statistics to model channel correlations. Instead, this method employs global second-order pooling (GSoP) to enhance the collection of global information [61]. The specific calculation process of this method is presented below:(7)$$\begin{array}{c} S^{low}=\sigma \left(\omega_{2}RC\left(Cov\left(f_{v}^{low}\right)\right)\right) \end{array}$$
(8)$$\begin{array}{c} f_{v}^{low^{\prime }}=S^{low}⊙f_{v}^{low} \end{array}$$

Cov(·) represents the calculation of the covariance matrix; RC(·) means row-wise convolution, $s^{low}\in R^{C_{1}\times 1\times 1}$.

To promote feature consistency in the channel dimension, the high-level and global semantic features were computed using the same method:(9)$$\begin{array}{c} f_{v}^{high^{\prime }}=\sigma \left(W_{2}RC\left(Cov\left(f_{v}^{high}\right)\right)\right)⊙f_{v}^{high} \end{array}$$
(10)$$\begin{array}{c} f_{v}^{{\mathrm{glb}}^{\prime }}=\sigma \left(W_{2}RC\left(Cov\left(f_{v}^{\mathrm{glb}}\right)\right)\right)⊙f_{v}^{\mathrm{glb}} \end{array}$$

It is important to note that the computations described above did not alter the dimensions of the feature levels.

#### 3.2.3. Self-Attention Based on the Transformer Framework

The multi-level image features obtained from the channel attention module were transformed into image region feature sequences suitable for processing via a transformer encoder. This transformation was accomplished through patch embedding, which essentially encompasses a particular convolutional operation.
(11)$$\begin{array}{c} f_{v}^{low^{\prime \prime }}=Conv1\left(f_{v}^{low^{\prime }}\right) \end{array}$$
(12)$$\begin{array}{c} f_{v}^{high^{\prime \prime }}=Conv2\left(f_{v}^{high^{\prime }}\right) \end{array}$$
(13)$$\begin{array}{c} f_{v}^{glb^{\prime \prime }}=Conv3\left(f_{v}^{glb^{\prime }}\right) \end{array}$$
(14)$$\begin{array}{c} f_{v}=Cat\left(f_{v}^{low^{\prime \prime }},f_{v}^{high^{\prime \prime }},f_{v}^{{\mathrm{glb}}^{\prime \prime }}\right) \end{array}$$

From the low-level to global semantic features, the used convolution kernel sizes were reduced to 1×1 gradually. In simpler terms, the approach aimed to capture large-scale receptive fields from both low-level and high-level features while preserving the comprehensive global semantic features. The objectives of this approach were twofold: (1) to minimize the number of image regions to balance the subsequent computational workload, and (2) to mitigate information loss of the global semantic features, which possess the highest semantic discrimination capability. Using the above computations, we obtained the multi-scale region feature sequences $f_{v}$ ($f_{v}\in R^{l\times h}$, where *l* represents the total number of regions after multi-scale features were connected; *h* represents the characteristic length of a single region).

After that, $f_{v}$ was sent to three stacked transformer encoders to enhance the feature consistency through vision self-attention. During this process, emphasis was placed on reinforcing prominent features and examining the relationship between objects of different scales within the image. The computation process is documented below, as it mirrors the standard encoding process of the transformer:(15)$$\begin{array}{c} {f_{v}}^{{}^{\prime }}=\left\{v_{1},v_{2},\ldots \ldots ,v_{l}\right\}=Transformer\left(f_{v},\theta_{trans}\right) \end{array}$$

*v* represents different tokens in the feature sequences and practically represents different regions of the image in different scales: $v\in R^{l}$.

### 3.3. Fine-Grained Semantic Alignment Network

Through a cross-attention computation in this network, the feature sequences of the multi-scale image and text achieved precise semantic alignment. Given that image features typically contain more information than text features, the text was employed as a query to guide the alignment of image features with respect to the text features (Figure 5).

To begin, the region features and word features were individually mapped using three fully connected layers. $\forall v_{j}\in f_{v}^{\prime }$ and $\forall e_{k}\in f_{t}$ are followed in the following formulas.
(16)$$\begin{array}{c} e_{k}^{que}=\omega_{que}e_{k} \end{array}$$
(17)$$\begin{array}{c} v_{j}^{key}=\omega_{key}v_{j} \end{array}$$
(18)$$\begin{array}{c} v_{j}^{val}=\omega_{val}v_{j} \end{array}$$

$\omega_{que}$, $\omega_{key}$, and $\omega_{val}$ represent the parameter matrices of query, value, and key, respectively, and all adopt the dimension of h×h. Thereby, the attention of an image region can be expressed below:(19)$$\begin{array}{c} S_{kj}=\frac{v_{j}^{key^{T}}e_{k}^{que}}{∥v_{j}^{key}∥∥e_{k}^{que}∥},k\in \left[1,p+1\right],j\in \left[1,l\right] \end{array}$$
(20)$$\begin{array}{c} \bar{S_{kj}}=\frac{\lambda S_{kj}}{\sqrt{\sum_{j=1}^{l}{\left(\lambda S_{kj}\right)}^{2}}} \end{array}$$
(21)$$\begin{array}{c} \lambda =\frac{u}{1+e^{\left(r-h\right)}} \end{array}$$

The normalization operation of softmax is denoted by Equation (Equation 20). Hence, $\bar{S_{kj}}$ represents the normalized weight of the j-th image region obtained under the guidance of the k-th word in a text; λ in Equation (Equation 20) represents the temperature coefficient at this time. The function of temperature coefficient λ is to adjust the saliency of image regions by scaling attention scores. Elevated temperatures led to irregular distributions of significance among regions, potentially aiding the learning of difficult negative examples. Conversely, lower temperatures had the opposite effect. Equation (Equation 21) shows the process of solving the temperature coefficient: h denotes the rounds of training; u is the ultimate constant to which the function converges; r denotes at which round of training the median point should be achieved. The variation of λ with the rounds of training is shown in Figure 6.

Based on the normalized weights, the image feature representation $f_{v}^{k}$ guided by the k-th word can be calculated and expressed below:(22)$$\begin{array}{c} f_{v}^{k}=\sum_{j=1}^{l}\bar{S_{kj}}v_{j} \end{array}$$

The similarity between text t and image m can be expressed as the sum of similarities between the words and their corresponding image features, where each word guides the generation of the respective image feature.
(23)$$\begin{array}{c} S\left(m,t\right)=\frac{1}{p+1}\sum_{k=1}^{P+1}\frac{f{{}_{v}^{k}}^{{}^{T}}e_{k}}{∥f_{v}^{k}∥∥e_{k}∥},k\in \left[1,p+1\right] \end{array}$$

### 3.4. Target Function

In this study, we employed the bidirectional triplet ranking loss function proposed in [62] as a constraint to encourage close proximity between matching image and text pairs in a shared feature space, while ensuring separation between unmatched image and text pairs. The formulation of this loss function is as follows:(24)$$\begin{array}{c} L\left(M,T\right)=\sum_{i=1}^{b}\left\{\max\left[\beta +S\left(m_{i},t_{i}^{\prime }\right)-S\left(m_{i},t_{i}\right),0\right]+\max\left[\beta +S\left(m_{i}^{\prime },t_{i}\right)-S\left(m_{i},t_{i}\right),0\right]\right\} \end{array}$$
where β represents the margin and represents the preset cosine distance between positive and negative examples. The selection of an appropriate margin is crucial for achieving desired modeling outcomes. Since the small-batch training strategy is often used in training to update network parameters stably, $m_{i}^{\prime }$ only represents all negative image samples in the same batch. Furthermore, $t_{i}^{\prime }$ represents all negative text samples in the same batch, and *b* represents the size of the minibatch. In training, L(M,T) is minimized to obtain the optimum model parameters.

## 4. Experimental Results

To assess the effectiveness of the proposed FAAMI approach, we carried out comprehensive experiments on two well-established cross-modal public datasets in remote sensing, specifically RSICD and RSITMD. These datasets are recognized for their extensive size and precise annotations. The experiments can be classified into two categories: basic experiments and ablation study experiments. In the basic experiments, we compared the performance metrics of FAAMI with the state-of-the-art non-fusion coding model, which is the most advanced model of its kind, on both datasets. The results of this comparison demonstrate the effectiveness of FAAMI. The ablation study experiments, presented in another part of this section, involved conducting numerous contrastive experiments by replacing and dismantling different modules while controlling for variables. These contrastive results provide strong evidence supporting the rationality and effectiveness of the innovative aspects proposed in this paper.

### 4.1. Dataset and Assessment Indices

(1) RSICD: The dataset used in this study is a comprehensive RS image captioning dataset developed by Lu et al. [63]. It comprises a total of 10,921 RS images, each belonging to 1 of 30 categories, such as airports and industrial parks. Each image in the dataset is of size 224×224 pixels and is accompanied by descriptive text consisting of five sentences.

(2) RSITMD: The dataset used in this study is a fine-grained semantic annotation dataset introduced by Yuan et al. [18]. It consists of images from 32 different categories of scenarios. Each RS image in the dataset has a size of 256×256 pixels and is associated with a five-sentence description text as well as one to five fine-grained keywords.

(3) Assessment indices: In this study, two metrics were employed to evaluate the effectiveness of the model. The first metric is Recall at K (R@K, K = 1, 5, 10), which is commonly used in image detection tasks and indicates the percentage of positive examples among the top K images (text). The second metric is mR, as reported by Huang et al. [64], which represents the average of six recall rates.

### 4.2. Experimental Settings

The datasets were divided into training, validation, and testing sets according to the provided division methods. The quantitative proportions of the three sets were set to 8:1:1. All experiments were performed on an Ubuntu 18.4 system using four NVIDIA-3090 graphics cards for training. The training process lasted for 30 epochs, with a minibatch size of 16. The value of β was 0.25. The initial learning rate was set to 0.00003 and smoothly decreased following a cosine schedule during training.

### 4.3. Basic Experiments

#### 4.3.1. Basic Experiments on RSICD

For performance comparison, we selected three recent common feature space models, namely AMFMN [18] and its three variants (AMFMN-soft, AMFMN-fusion, AMFMN-sim), HyperMatch [30], and CMFM-Net [24], as baseline models based on RSICD. The reasons for selecting them are as follows: Firstly, they both belong to the common feature space approach and address the multi-scale problem. Secondly, they are both advanced models proposed within the past year. Thirdly, similar to FAAMI, they have undergone experimental evaluation on the same dataset under identical conditions. The particular methods utilized by these three models have been presented in the introduction and will not be restated here. Additionally, we included SCAN (SCAN-i2t and SCAN-t2i) [65], a well-known cross-modal model for natural images, for comparison. The proposed FAAMI shares a similar concept with SCAN but addresses multi-scale challenges more comprehensively.

According to the results presented in Table 1, the proposed FAAMI outperformed all the baseline models across all six indices. In text retrieval, FAAMI achieved higher R@1 and R@5 values by 3.3% and 2.62%, respectively, compared to the best-performing baseline model. The R@10 value was only 0.13% lower, indicating a minimal difference. For image retrieval, FAAMI significantly surpassed all baseline models, with R@1, R@5, and R@10 values being 2.03%, 5.22%, and 7.55% higher than the best baseline model, respectively. When evaluating the mR metric, which assesses the overall performance of the models, FAAMI outperformed the best baseline model by 3.43%. These results corroborate the effectiveness of the proposed FAAMI model.

#### 4.3.2. Basic Experiments on RSITMD

Considering the more detailed text annotation provided by RSITMD, we expected that the proposed FAAMI model would demonstrate better performance on this dataset. To provide a baseline for comparison, we included a fusion coding model called FBCLM [28] in our evaluation.

As shown in Table 2, the FAAMI model outperformed the baseline models in all seven evaluation metrics, and also achieved higher results compared to FBCLM to some extent. It is worth noting that FBCLM employs a large-scale semantic interaction network (a three-layer cross-attention network), which results in higher retrieval costs compared to FAAMI. These findings suggest that FAAMI provides a more streamlined solution with a shorter retrieval response time and exhibits superior performance on the RSITMD dataset. Specifically, in text retrieval, FAAMI achieved improvements of 3.31% in R@1, 5.09% in R@5, and 3% in R@10 compared to the optimum baseline model. In image retrieval, FAAMI surpassed the optimum baseline model by 2.52% in R@1, 5.38% in R@5, and 2.02% in R@10. Moreover, the mR metric of FAAMI outperformed models of the same kind by 6.27% and surpassed the fusion coding model by 3.55%. These results collectively support the superiority of the proposed FAAMI model.

#### 4.3.3. Search Results

Using the RSITMD dataset, we evaluated the retrieval capabilities of the models. Figure 7 illustrates the image search results of FAAMI based on text queries. Correctly matched images are highlighted with green frames, while incorrect matches are indicated with red frames. The numbers beneath the frames denote the ranking of the results. Likewise, Figure 8 illustrates the text search results of FAAMI using image queries, where frame colors signify correctness and numbers indicate rankings. It is crucial to emphasize that the accuracy of the retrieval results was rigorously evaluated in accordance with the dataset annotations. For example, in Query.A, the first image is highly relevant to the text, while distinguishing between images (1) and (2) is challenging. In Query.B, the first five recalled images share significant visual similarity, with differentiation relying on factors such as multi-scale target-associated words like “interspersed” and detailed descriptions like “red and ultramarine flats”. Analyzing the text-based search results depicted in Figure 8, we observed that most recalled texts exhibit similar semantics, with subtle differences in quantifiers or target-related words. These retrieval results indicate that the proposed FAAMI model possesses excellent capabilities in fine-grained semantic discrimination and multi-scale target relationship modeling.

### 4.4. Ablation Study Experiments

To thoroughly examine the contributions of key modules and parameters in FAAMI, we conducted a series of ablation study experiments across four categories: (a) We replaced the backbone network for image coding to confirm the rationality of selecting DetNet. (b) We deleted or replaced the modules responsible for constructing multi-scale image representations to assess the effectiveness of the multi-scale approach employed in FAAMI. (c) We compared the text-to-image and image-to-text cross-attention guidance methods to validate the rationale of FAAMI under multi-scale image representation conditions. (d) We adjusted the margin setting in the loss function to assess how the model’s performance varied with different margins. These ablation study experiments were devised to thoroughly examine the distinct contributions of these critical modules or parameters in FAAMI.

#### 4.4.1. Different Backbone Networks of Image Coding

To evaluate the adaptability of different backbone networks to the proposed FAAMI model, we conducted experiments by substituting DetNet-59 with ResNet-50 and ResNet-152 in the absence of multi-scale image representations. Three indices, namely mR (mean Recall), FLOPs (computation complexity), and time (training time per epoch), were used to compare the performance of these models.

The experimental results are presented in Table 3, allowing for a comprehensive assessment of the impact of different backbone networks of image coding on cross-modal RS image retrieval. The FLOPs (computation complexity) provided in Table 3 represent the values reported in the original papers, although the removal of classifiers or target detectors would result in lower actual FLOPs. Nonetheless, the ranking of the models remained consistent. In the left part of Table 3, it can be observed that DetNet had slightly higher FLOPs compared to ResNet-50, while both DetNet and ResNet-50 had significantly lower FLOPs than ResNet-152. This observation was further supported by the actual operation times, with ResNet-50 exhibiting 4.06% and 16.11% shorter operation times compared to the other two models, respectively. In terms of model precision on the RSICD dataset, the ranking aligned with the computation complexity. However, the difference in precision between DetNet and ResNet-50 was marginal, and the precision of ResNet-152 was only 1.47% higher than the other two models, which was unexpected. It was initially anticipated that DetNet, with its higher feature resolution, would be more advantageous for preserving small-target information and would exhibit higher precision. The small discrepancy of 0.09 in fine-grained semantic interaction precision between DetNet and ResNet-50 raised questions. To investigate whether the observed results were influenced by the semantic annotation granularity in the dataset, we conducted the same experiments on the RSITMD dataset.

The outcomes of the repeated experiments on RSITMD can be found in the right section of Table 3. Since RSITMD has a finer annotation granularity compared to RSICD, the results of the repeated experiments aligned with our expectations. In terms of model precision, DetNet, serving as the backbone network of FAAMI, demonstrated nearly identical performance to ResNet-152, with only a slight 0.24% difference. Additionally, DetNet surpassed ResNet-50 by 1.03%. Therefore, taking into account its lightweight characteristics and precision, utilizing DetNet as the image coding backbone network in the proposed FAAMI model is a suitable choice.

#### 4.4.2. Different Multi-Scale Feature Construction Methods and Feature Consistency Enhancement Modules

In this set of experiments, six variations of FAAMI were developed to investigate the effects of various methods for multi-scale feature construction and FCEM on the performance of the model.

(1) FAAMI-No MS: this model only adopted the final feature map of DetNet without constructing multi-scale representations.

(2) FAAMI-with SPP: The spatial pyramid pooling method (SPP) was used as a replacement for the cross-layer feature connection method in FAAMI. In this modified model, multi-scale feature representations were constructed using the final feature map of DetNet.

(3) FAAMI-No FCEM: In this model, the feature consistency enhancement module was excluded from FAAMI.

(4) FAAMI-No GSoP: In this model, the feature consistency enhancement module had its channel attention modules removed.

(5) FAAMI-No TF: In this model, the feature consistency enhancement module had its transformer encoders removed.

(6) FAAMI-with SPP & No FCEM: This model, referred to as “FAAMI-with SPP,” does not include the feature consistency enhancement module.

The experimental results of the six variant models and the original model on the RSICD and RSITMD datasets are presented in Table 4 and Table 5. The training time for both datasets followed a similar trend. Regardless of the presence of the FCEM, the SPP method showed a slightly shorter training time (approximately 2%) compared to the cross-layer feature connection method. This small difference can be attributed to the cascading operation of features. The GSoP module within the FCEM had the most notable impact on computation overhead. It increased the computation overhead by approximately 30% due to the computation of higher-order statistics and the inter-channel covariance matrix, which demanded a considerable amount of time. On the other hand, the image self-attention module had a relatively minor impact (8–10%) on computation overhead. Additionally, when the FCEM was added to both the FAAMI with SPP and the FAAMI with cross-layer connection method simultaneously, the computation overheads of the two models increased to different extents (14% and 33.6%, respectively). This discrepancy can be attributed to the GSoP processing of low-level, high-level, and global semantic features in the FAAMI with cross-layer connection method, resulting in larger differences in time overhead between the two models.

Regarding model precision, both the FAAMI with cross-layer connection method and the FAAMI with SPP demonstrated effectiveness. Compared to the FAAMI-No MS, the mRs of these two models improved by 1.76% and 0.72% over the RSICD dataset, and by 2.33% and 1.23% over the RSITMD dataset, respectively. These results highlight the importance of constructing multi-scale feature representations for cross-modal semantic alignment. The comparative experiments that deconstructed the FCEM demonstrated its role in enhancing the accuracy of both FAAMI models, employing the cross-layer connection method and the SPP, albeit to different extents. Following the inclusion of the FCEM in both models, the mRs improved by 0.61% and 1.7% for the RSICD dataset, and by 0.53% and 1.72% for the RSITMD dataset, respectively. These findings indicate that the impact of the FCEM was more pronounced in the FAAMI with cross-layer connection method, aligning with its original design intention. Furthermore, we conducted contrastive experiments by individually removing the GSoP and self-attention module from the FCEM. The four sets of experimental results obtained via the two datasets demonstrate that each module (GSoP or self-attention) in the FCEM contributed to improving model precision. However, the exact role of each module could not be definitively determined based on the existing experimental results. The differences observed between the datasets and the slight variation of only 0.2% raise the possibility of interference from random errors and dataset variations, which require further investigation.

#### 4.4.3. Different Semantic Interaction Methods

According to the computation rules of cross-attention networks, searching can be performed based on either image features or text features. In the proposed FAAMI, we hypothesized that, with multi-scale visual representations as a prerequisite, the inclusion of multi-scale target-related word information in the text would serve as crucial cues for aggregating multi-scale visual information. Without this, individual image regions would struggle to guide the aggregation of word information from the text. Therefore, we adopted a text-to-image semantic interaction approach in this study. To validate the validity of this idea, we conducted two sets of contrastive experiments (group A and group B) on the RSITMD dataset. In group A experiments, we compared the mRs of different semantic guidance methods when multi-scale image representations were available. In group B experiments, we compared the mRs of different semantic guidance methods in the absence of multi-scale image representations. The results are depicted in Figure 9.

Based on a cross-contrast analysis of the two groups of experimental results, the following conclusions were drawn:

(1) In the context of multi-scale image representations, there was a significant difference (2.42%) in precision between the two semantic guidance methods (text-to-image and image-to-text), indicating that the benefits of multi-scale enhancement were completely offset by the choice of semantic guidance method.

(2) Without multi-scale image representations, the precision difference between the two semantic guidance methods was nearly negligible.

(3) When comparing the data from group A and group B, it was noticed that the precision difference between the two methods was relatively minor. This suggests that integrating multi-scale visual information did not offer substantial advantages to the model when employing an image to guide the aggregation of information in the text.

Primarily, these experimental findings align with the aforementioned inference. The FAAMI model exhibits sensitivity to variations in the semantic interaction method primarily due to the incorporation of multi-scale vision information. This is evident as the FAAMI model without the multi-scale module demonstrated almost indistinguishable precisions in both semantic interaction methods.

#### 4.4.4. Different Margins

The margin refers to the predefined cosine distance between positive and negative examples within the models. Optimal margins may vary across datasets and models, as an excessively large margin can disrupt the balance of feature distribution, while a margin that is too small may hinder effective discrimination of negative examples. Based on RSCID, we compared the mRs of FAAMI when the margin was $\left\{\left.\beta \right|0.2,0.25,0.3,0.35,0.4\right\}$ to understand the impact of margin on the model’s performance.

As shown in Figure 10, the maximum mR difference between the models with various margins reached 1.39% when the margins β were between 0.25 and 0.4. When β≤0.35, the model’s performance change was uniform. Disregarding random errors, the mR difference was constrained to a maximum of 0.9%. Based on the aforementioned experimental results, a preliminary inference can be drawn: the proposed FAAMI model exhibits a stable performance within a specific margin range. In contrast, its performance is significantly impacted when the margin exceeds this range. This conclusion is logically understandable. As there are many soft negative examples (the similarity between texts is high) in the RSICD, the model with over-large β may distinguish the image–text pairs with similar semantics excessively, which may affect the convergence of the model.

## 5. Discussion

During the experiments, an intriguing phenomenon was observed. Specifically, two versions of pre-trained models, namely Bert-large and Bert-base, were utilized as text encoders within the cross-modal network. Bert-large comprises 24 Bert encoders and 330 million parameters, whereas Bert-base consists of only 12 Bert encoders and 110 million parameters. As a result, Bert-large often outperforms Bert-base in various natural language processing tasks, such as text sentiment classification. Additionally, owing to the elevated feature dimensionality of Bert-large encoding, it was expected that this feature could assist in minimizing information loss during the patch embedding process and improve the model’s precision. However, in contrast to expectations, the experimental results revealed a notable deterioration in FAAMI’s performance when employing Bert-large. Specifically, on the RSICD and RSITM datasets, the mRs values of the FAAMI variant with Bert-large decreased by 9.65% and 5.66%, respectively.

In order to investigate whether other factors contribute to the aforementioned anomalous results, control variable experiments were conducted on the settings of learning rate and learning strategy. As presented in Figure 11, there are three groups of experiments (group A, group B, and group C) with three initial learning rates ($3\times 10^{-5}$, $2\times 10^{-5}$, and $1\times 10^{-5}$); but in the training process, the learning rates dropped to $1\times 10^{-6}$, though still in a cosine manner. During the experiments conducted on group D, a learning strategy known as “warming up” was employed, whereby the learning rate gradually increased and then decreased exponentially. As indicated by the experimental findings, utilizing different learning strategies resulted in a certain degree of improvement in the model’s performance. The mR of group D also reached 19.57%, albeit still considerably lower than the FAAMI variant with Bert-base.

Based on the aforementioned experimental results, it is hypothesized that the observed phenomenon may be attributed to substantial disparities in the number of parameters and the network depth between the image encoder and text encoder. In the FAAMI variant with Bert-base, the image and text encoders comprise approximately 110 million parameters and have nearly identical network depths. However, in the FAAMI variant with Bert-large, the text encoder has 330 million parameters, whereas the image encoder has only 130 million parameters, resulting in a doubled difference in network depths between the two encoders. Such disparities could potentially impact the training of single-modal encoders, leading to substantial differences in semantic distribution between the image and text data. Thus, maintaining balance in the single-modal encoders may aid in preserving the model’s performance in cross-modal RS image retrieval. Additionally, if the overall parameter scale of the model is increased, there is a heightened risk of overfitting and partial convergence of the FAAMI on the datasets. The proposed FAAMI model might be susceptible to the impact of this factor. It is crucial to emphasize that the aforementioned suspicion is grounded in logical reasoning and necessitates further empirical validation in future research endeavors.

## 6. Conclusions

In this paper, we propose a novel FAAMI method that leverages cross-layer feature connections to extract multi-scale features. This approach enhances the fine-grained semantic alignment process by incorporating a multi-scale perspective, enabling the network to model complex semantic correlations between multi-scale objects and textual words. To address the challenge of semantic inconsistency across layers, the model integrates a feature consistency enhancement module that calculates attention across channel and spatial dimensions. Importantly, to ensure practical usability for retrieval tasks, the model employs a shallow cross-attention network (one Bert encoder) to attain fine-grained cross-modal semantic alignment. Extensive experiments conducted on two public datasets demonstrate the superiority of the proposed FAAMI over other state-of-the-art baseline models.

However, the FAAMI method also has some limitations. Traditional CNN frameworks used to mine multi-scale features suffer from issues such as input redundancy, network redundancy, and cross-layer inconsistency. Although the FCEM module employed in this study partially mitigates feature inconsistency. It introduces additional computational overhead to the network. Consequently, future research may focus on developing a cross-modal RS image retrieval framework that employs the VIT as the vision encoder to extract multi-scale visual representations within the attention mechanism. This approach aims to make the model more concise and efficient while maintaining the same level of precision. Furthermore, the study will explore the impact of balancing bimodal encoders and mining self-supervised supervisory signals. 

## Figures and Tables

**Figure 1 sensors-23-08437-f001:**
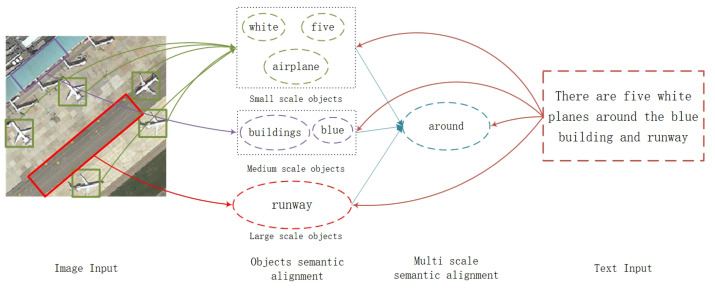
Schematic diagram of fine-grained semantic interaction specific to multi-scale enhancement. As shown in the figure, there are multiple targets with a large-scale difference, and there are semantic associations among these targets. Therefore, in addition to mining multi-scale target features, it is also necessary to consider their precise semantic alignment to the corresponding words in the text.

**Figure 2 sensors-23-08437-f002:**
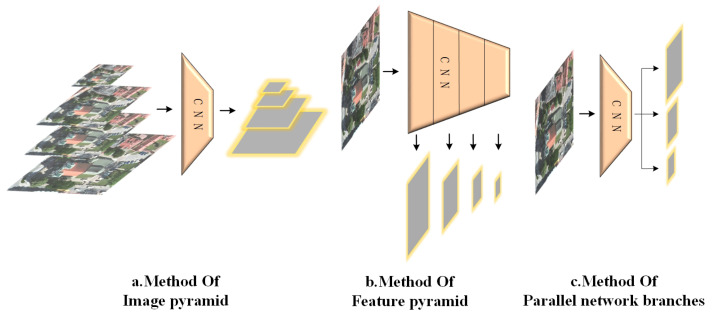
Schematic diagram of the three multi-scale feature construction methods.

**Figure 3 sensors-23-08437-f003:**
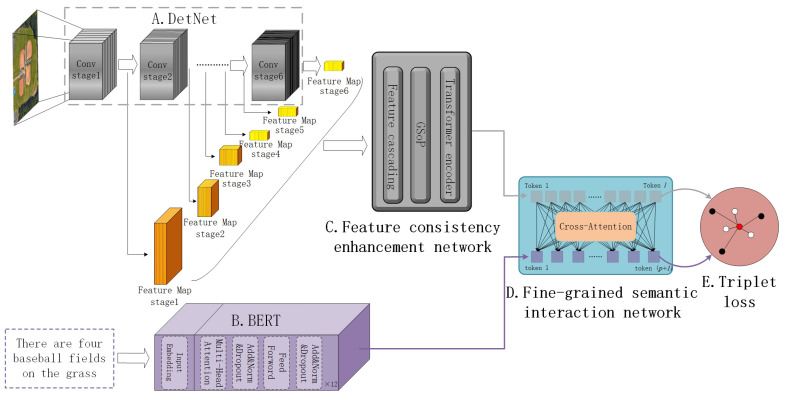
The overall structure of the network. (Module A is an image encoder; module B is a text encoder; module C represents a feature consistency enhancement network; module D is a fine-grained semantic interaction network constructed using the cross-attention layers; module E represents a triplet loss and is used to guide the final network training).

**Figure 4 sensors-23-08437-f004:**
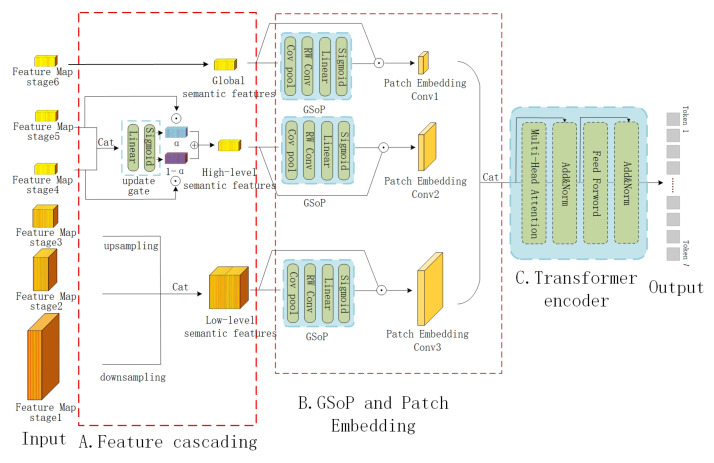
Detailed structure of the FCEM. The FCEM involves three modules: Module A performs the cascaded computation of cross-layer features, resulting in global, high-level, and low-level semantic features. Module B enhances feature consistency in the channel dimension using a GSoP module. Module C is a vision self-attention module composed of transformer encoders, which filters out redundant information and enhances feature consistency in the spatial dimension.

**Figure 5 sensors-23-08437-f005:**
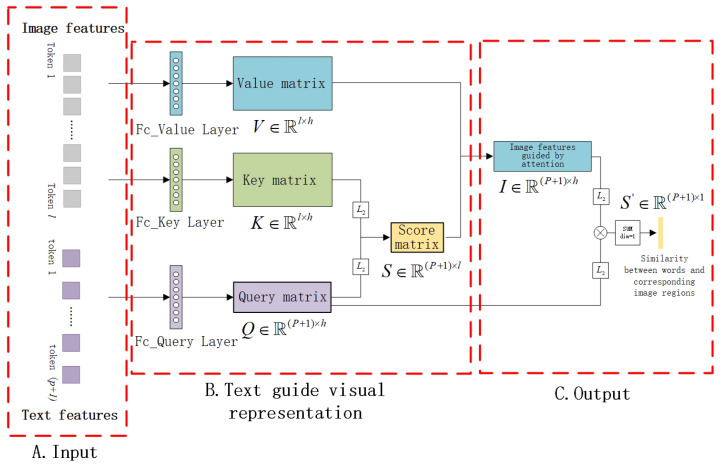
Structure of the fine-grained semantic interaction network. The main procedure of this network involves cross-attention computation utilizing text as a basis. Each word autonomously guides the image to generate feature representations. Ultimately, the average similarity of all word–image feature pairs is employed to represent the overall similarity between the text and image.

**Figure 6 sensors-23-08437-f006:**
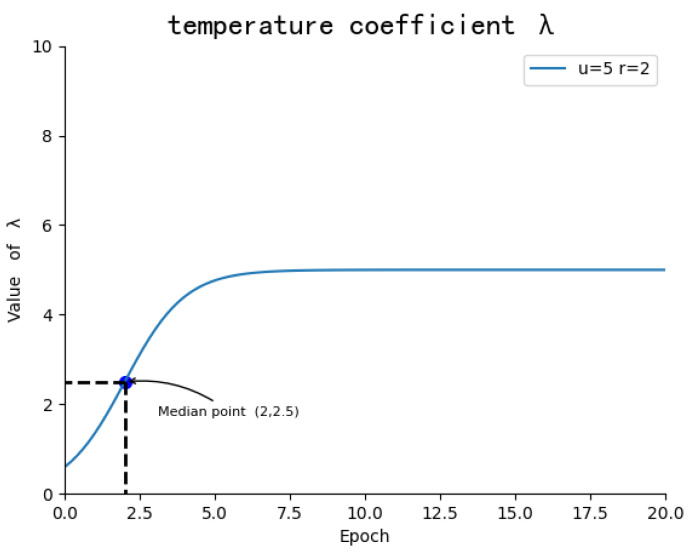
Graph of the temperature coefficient λ, with the values of the standard model’s parameters at the upper-right corner.

**Figure 7 sensors-23-08437-f007:**
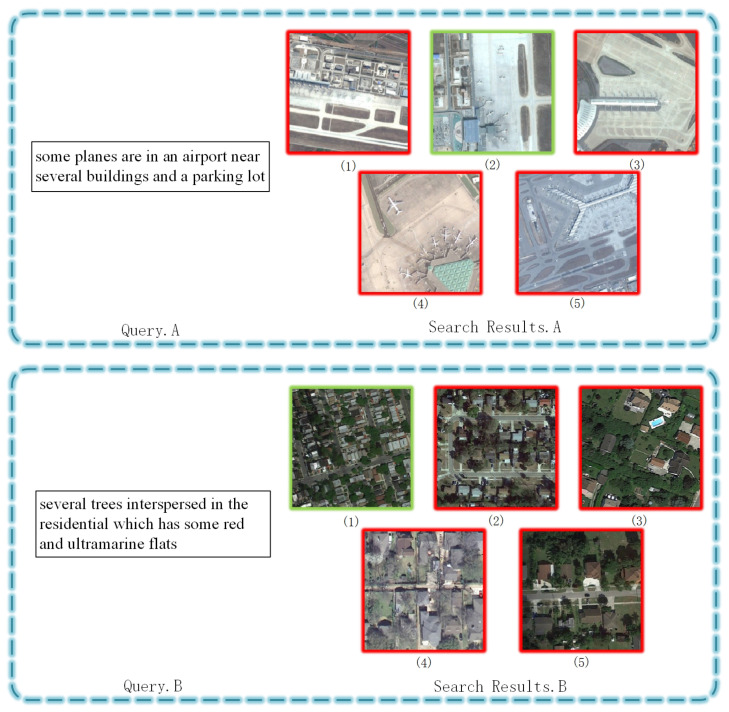
The text-based image search results of FAAMI over RSITMD (R@5). The green frame represents the positive example of images matching with the text.

**Figure 8 sensors-23-08437-f008:**
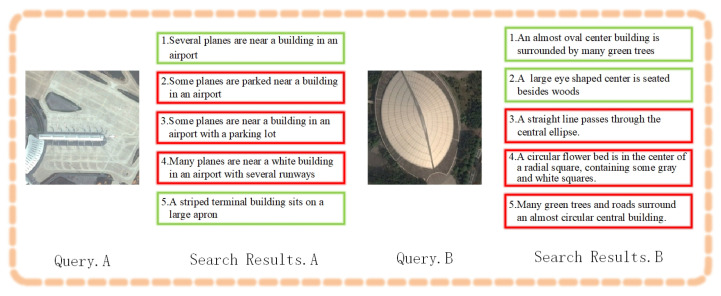
The image-based text search results of FAAMI over RSITMD (R@5).

**Figure 9 sensors-23-08437-f009:**
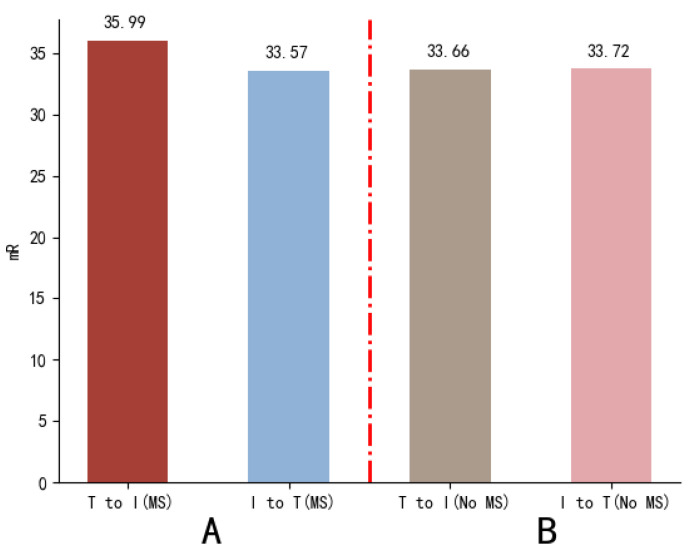
Results of the experiments in different semantic interaction methods. The red dotted line indicates the boundary, with the left part displaying the experimental results of group A and the right part representing the results of group B.

**Figure 10 sensors-23-08437-f010:**
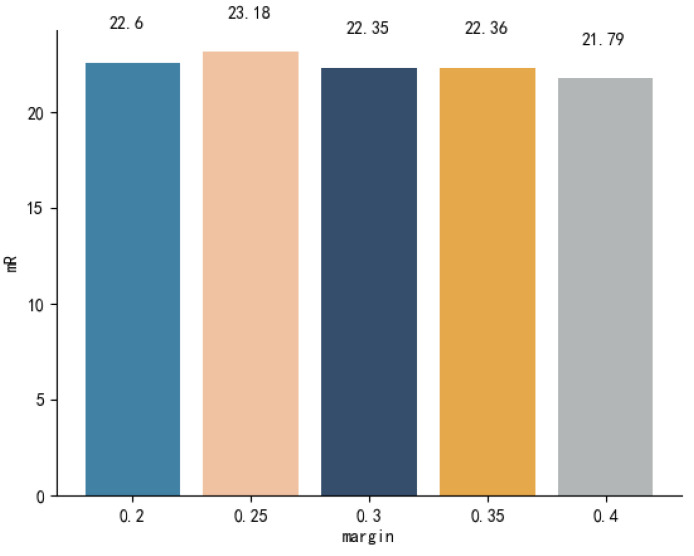
Performance of the models with different margins.

**Figure 11 sensors-23-08437-f011:**
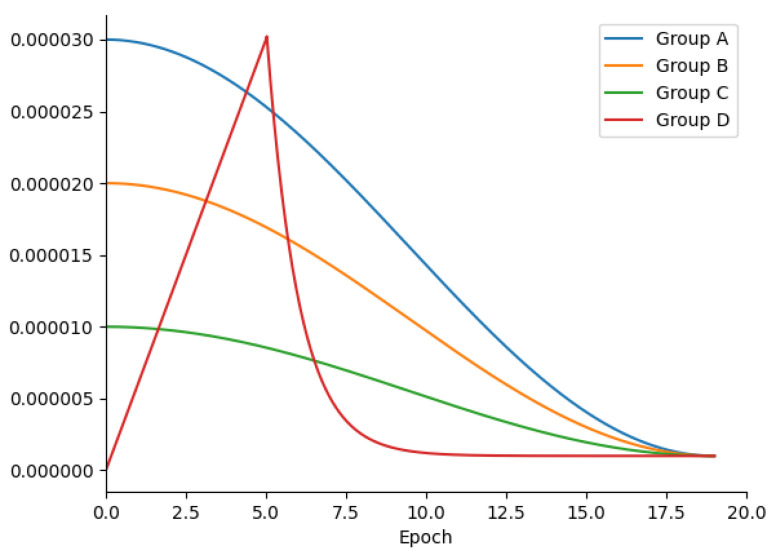
Schematic diagram of the learning strategies used in the four groups of experiments.

**Table 1 sensors-23-08437-t001:** Comparison of results of all models over the dataset RSICD.

Method	Text Retrieval			Image Retrieval			
	R@1	R@5	R@10	R@1	R@5	R@10	mR
SCAN-t2i	4.39	10.90	17.64	3.91	16.20	26.49	13.25
SCAN-i2t	5.85	12.89	19.84	3.71	16.40	26.73	14.23
AMFMN-soft	5.05	14.53	21.57	5.05	19.74	31.04	16.02
AMFMN-fusion	5.39	15.08	23.40	4.90	18.28	31.44	16.42
AMFMN-sim	5.21	14.72	21.57	4.08	17.00	30.60	15.53
CMFM-Net	5.40	18.66	28.55	5.31	18.57	30.03	17.75
HyperMatch	7.14	20.04	**31.02**	6.08	20.37	33.82	19.75
FAAMI	**10.44**	**22.66**	30.89	**8.11**	**25.59**	**41.37**	**23.18**

**Table 2 sensors-23-08437-t002:** Comparison of results of all models over the dataset RSITMD.

Method	Text Retrieval			Image Retrieval			
	R@1	R@5	R@10	R@1	R@5	R@10	mR
SCAN-t2i	10.18	28.53	38.49	10.10	28.98	43.53	26.64
SCAN-i2t	11.06	25.88	39.38	9.82	29.38	42.12	26.28
AMFMN-soft	11.06	25.88	39.82	9.82	33.94	51.90	28.74
AMFMN-fusion	11.06	29.20	38.72	9.96	34.03	52.96	29.32
AMFMN-sim	10.63	24.78	41.81	11.51	34.69	54.87	29.72
CMFM-Net	10.84	28.76	40.04	10.00	32.83	47.21	28.28
HyperMatch	11.73	28.10	38.05	9.16	32.31	46.64	27.66
FBCLM	12.84	30.53	45.89	10.44	37.01	57.94	32.44
FAAMI	**16.15**	**35.62**	**48.89**	**12.96**	**42.39**	**59.96**	**35.99**

**Table 3 sensors-23-08437-t003:** Comparison of the experimental results on different backbone networks of image coding.

RSICD				RSITMD			
Method	mR	Time	FLOPs	Method	mR	Time	FLOPs
FAAMI–Res50	21.33	**602 s**	**3.8** × 10^9^	FAAMI–Res50	32.63	**269 s**	3.8 × 10^9^
FAAMI–Res152	**22.89**	699 s	$11.3\times 10^{9}$	FAAMI–Res152	**34.06**	323 s	$11.3\times 10^{9}$
FAAMI–DetNet	21.42	626 s	$4.8\times 10^{9}$	FAAMI–DetNet	33.66	284 s	$4.8\times 10^{9}$

**Table 4 sensors-23-08437-t004:** Comparison of experimental results on models constructed with different multi-scale feature construction methods on the RSICD dataset.

RSICD								
Method	Text Retrieval			Image Retrieval			mR	Time
	R@1	R@5	R@10	R@1	R@5	R@10		
No MS	8.70	21.11	30.16	6.80	23.06	38.70	21.42	626 s
With SPP	9.52	21.66	28.79	7.13	25.56	40.16	22.14	675 s
No FCEM	8.07	21.12	28.98	6.87	24.26	39.60	21.48	603 s
No GSoP	8.88	20.92	30.26	6.38	24.72	41.15	22.05	664 s
No TF	9.34	21.93	30.71	7.06	24.33	39.91	22.21	793 s
with SPP & No FCEM	8.8	20.93	29.61	6.16	24.24	39.45	21.53	**592 s**
FAAMI	**10.44**	**22.66**	**30.89**	**8.11**	**25.59**	**41.37**	**23.18**	809 s

**Table 5 sensors-23-08437-t005:** Comparison of the experimental results on the models constructed with different multi-scale feature construction methods on RSITMD dataset.

RSITMD								
Method	Text Retrieval			Image Retrieval			mR	Time
	R@1	R@5	R@10	R@1	R@5	R@10		
No MS	13.71	30.42	48.23	11.58	40.13	57.88	33.66	284 s
With SPP	14.82	34.29	47.12	13.41	40.22	59.46	34.89	292 s
No FCEM	13.27	34.07	46.68	11.93	40.56	59.16	34.27	274 s
No GSoP	13.05	33.85	50.00	10.53	41.86	60.71	35.00	293 s
No TF	14.60	33.85	47.35	12.52	40.80	58.76	34.64	357 s
with SPP & No FCEM	11.95	33.41	48.45	11.81	41. 42	59.12	34.36	**253 s**
FAAMI	**16.15**	**35.62**	**48.89**	**12.96**	**42.39**	**59.96**	**35.99**	368 s

## Data Availability

Not applicable.

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
