# Peer review of "A Fine-Grained Semantic Alignment Method Specific to Aggregate Multi-Scale Information for Cross-Modal Remote Sensing Image Retrieval"

_sensors, 2023, doi:10.3390/s23208437_

Round 1

Reviewer 1 Report

This paper proposes a novel fine-grained semantic alignmen (FAAMI) method that leverages cross-layer feature connections to extract multi-scale features. This approach enhances the fine-grained semantic alignment process by incorporating a multi-scale perspective, enabling the network to model complex semantic correlations between multi-scale objects and textual words. To address the challenge of semantic inconsistency across layers, the model incorporates a feature consistency enhancement module that computes attention across channel and space dimensions. In order to ensure practical usability for retrieval tasks, the model adopts a shallow cross-attention network (one Bert encoder) to achieve fine-grained cross-modal semantic alignment. Full experiments conducted on two public datasets demonstrate the superiority of the proposed FAAMI over other state-of-the-art baseline models. The following suggestions might be beneficial to the article:

1. Please explain the meaning of the representation in Eq. 5. In addition, the symbol for the multiplication of elements in Eq. 4 is inconsistent with Fig. 5.

2. In subsection 3.2.2, GSoP acts on high-, low-, and glb-level feature maps to enhance global information. Why does this also promote feature consistency in the channel dimension? As described at the bottom of line 256.

3. It is difficult to understand Eqs. 20 and 21, as Eq. 20 lacks the explanation of lambda, and Eq. 21 lacks the explanation of u and r.

4. Please explain the meaning of theta_low in Eq. 7 and the meaning of g(h) in Fig. 6.

5. Please provide specific settings for the important parameters u, r and beta in subsection 4.2 of the experiment.

6. Lack of citations for the methods compared in subsection 4.3.1 of the experiment.

7. Comparison with the 2023 SOTA results is missing.

There are some typos in the paper. For example, the information below the title has initials that should be capitalized and spaces that are missing, "f_v^{mid}" would make more sense as "f_v^{high}" in line 251, "vrepresents" should be "v represents" in line 271.

Reviewer 2 Report

This paper proposed a new semantic alignment method for cross-modal remote sensing image retrieval, which is named as FAAMI. Generally, this paper is interesting and well organized. I would suggest this paper to be accepted after addressing the following problems.

1. The introduction part can be improved. More explanation should be added about why cross-modal retrieval is important for the community.

2. Moreover, the authors claimed that the retrieval methods can be classified into two categories, i.e., those based on a common feature space and fusion coding. One would expect that the common feature space method should be introduced clearly in the following part. However, the author introduced the second method firstly.

3. When abbreviations appear for the first time, please give the full name. I found that there are several abbreviations show up without full name.

4. Inconsistency between Fig.2 and Section 2. According to Fig.2, the authors should introduce Fig.2 a first, then Fig.2 b and Fig.2 c. However, the orders are not like this way. Moreover, does cross-layer feature connection correspond to Fg.2b?

5. Section 4.3.1, the comparison methods should be briefly introduced, with emphasis on why they are selected, what's the difference between these methods and the FAAMI.

Minor editing of English language required.

Reviewer 3 Report

Image retrieval is one of basic tasks of data processing, and embraces many variations and methods adapted therefor. The paper is devoted to one of such variations, namely image retrieval from domains of different scales and modalities. The authors developed an elaborate framework including several neural network architectures. The experiments include ablation study and are done with public domain databases, showing good potential of the method. The paper can be published with minor corrections: text in figures should be made bigger to be more readable.

Many times unnecessary dashes are inserted in the words, for instance "ad-dressed" in abstract, "se-mantic" in lines 113 and 114, "trans-former" in line 188 etc.

Round 2

Reviewer 1 Report

Please publish this article.

None.